# Could Vitamin D Be Effective in Prevention of Preeclampsia?

**DOI:** 10.3390/nu13113854

**Published:** 2021-10-28

**Authors:** Elżbieta Poniedziałek-Czajkowska, Radzisław Mierzyński

**Affiliations:** Chair and Department of Obstetrics and Perinatology, Medical University of Lublin, 20-954 Lublin, Poland; radek@bg.umlub.pl

**Keywords:** preeclampsia, vitamin D, prophylaxis

## Abstract

Prevention of preeclampsia (PE) remains one of the most significant problems in perinatal medicine. Due to the possible unpredictable course of hypertension in pregnancy, primarily PE and the high complication rate for the mother and fetus/newborn, it is urgent to offer pregnant women in high-risk groups effective methods of preventing the PE development or delaying its appearance. In addition, due to the association of PE with an increased risk of developing cardiovascular diseases (CVD) in later life, effective preeclampsia prevention could also be important in reducing their incidence. Ideal PE prophylaxis should target the pathogenetic changes leading to the development of PE and be safe for the mother and fetus, inexpensive and freely available. Currently, the only recognized method of PE prevention recommended by many institutions around the world is the use of a small dose of acetylsalicylic acid in pregnant women with risk factors. Unfortunately, some cases of PE are diagnosed in women without recognized risk factors and in those in whom prophylaxis with acetylsalicylic acid is not adequate. Hence, new drugs which would target pathogenetic elements in the development of preeclampsia are studied. Vitamin D (Vit D) seems to be a promising agent due to its beneficial effect on placental implantation, the immune system, and angiogenic factors. Studies published so far emphasize the relationship of its deficiency with the development of PE, but the data on the benefits of its supplementation to reduce the risk of PE are inconclusive. In the light of current research, the key issue is determining the protective concentration of Vit D in a pregnant woman. The study aims to present the possibility of using Vit D to prevent PE, emphasizing its impact on the pathogenetic elements of preeclampsia development.

## 1. Introduction

Preeclampsia belongs to the group of hypertensive diseases in pregnancy, which affect 8–10% of pregnant women. Chronic hypertension is observed in 0.9–1.5% of pregnancies, while gestational hypertension (GH) and preeclampsia (PE) could be diagnosed in 2–10% of pregnant women [1,2]. The PE incidence is estimated at approximately 1% of all pregnant women and 1.5% of primiparas [3]. Preeclampsia is one of the most important causes of maternal morbidity and mortality, mainly in developing countries. It is estimated that 16% of maternal deaths are PE-related. Preterm birth is the most common consequence of PE in developed countries [4,5].

Preeclampsia is a set of clinical symptoms that appears after the 20th week of pregnancy. It is a multi-organ disease characterized by hypertension and proteinuria, and in the absence of proteinuria—an impairment of the functions of the internal organs. According to the American College of Obstetricians and Gynecologists (ACOG) PE is defined by new-onset hypertension after the 20th week of gestation with systolic blood pressure ≥ 140 mmHg or diastolic blood pressure ≥ 90 mmHg, measured on two occasions at least 4 h apart, and proteinuria of ≥0.3 g per 24 h or ≥ 1+ proteinuria, detected by urine dipstick. PE could also be diagnosed in the absence of proteinuria when new-onset hypertension with new onset of any one of the following symptoms: thrombocytopenia (platelet count < 100,000/μL), renal insufficiency (serum creatinine concentration > 1.1 mg/dL or a doubling of the serum creatinine concentration in the absence of other renal diseases), impaired liver function (raised concentrations of liver transaminases to twice average concentrations), pulmonary edema, or cerebral or visual problems occur [2]. Another form of pregnancy-related hypertension is gestational hypertension or pregnancy-induced hypertension (PIH). It is recognized with systolic blood pressure ≥ 140 mmHg or diastolic blood pressure ≥ 90 mmHg appearing after the 20th week of pregnancy in previously healthy women, without proteinuria and other symptoms of multi-organ impairment typical of PE [2].

So far, the only causal treatment for PE is the delivery, and pharmacological management is symptomatic treatment only. Due to the possible unpredictable course of hypertension in pregnancy, primarily PE and the high complication rate for the mother and fetus/newborn, it is highly urgent to offer pregnant women from high-risk groups effective methods of preventing the development of this disease or delaying its appearance. In addition, due to the association of PE with an increased risk of developing cardiovascular diseases (CVD) in later life, effective prevention of preeclampsia could also be important in reducing their incidence [6].

Ideal PE prophylaxis should target the pathogenetic changes leading to the development of PE and be safe for the mother and fetus, inexpensive and freely available.

Currently, the only recognized method of PE prevention recommended by many institutions around the world is the use of a small dose of acetylsalicylic acid in pregnant women with risk factors. These include, among others, the age of the pregnant woman (<18 years and >40 years of age), first pregnancy, multiple pregnancy, pregnancy with a new partner, obesity, chronic hypertension, diabetes, chronic kidney diseases and autoimmune diseases (systemic lupus, antiphospholipid syndrome), and PE in the past [2,7,8,9].

Unfortunately, some cases of PE are diagnosed in women without recognized risk factors and in those in whom prophylaxis with acetylsalicylic acid is not adequate [10].

Hence the interest in other methods of preventing PE, which include antioxidants (vitamins C and E), calcium supplements, fish oil, nitric oxide supplements, nitric oxide donors, metformin, folic acid, statins, vitamins, weight loss, and physical activity, does not wane [11,12,13,14].

In recent years, much attention has been paid to the pleiotropic role of vitamin D (Vit D) in pregnancy as a substance with auto-, para-, and endocrine effects and the possibility of its use in PE prevention. Many studies published so far have indicated the importance of Vit D in fertilization, placental development, the course of pregnancy, and offspring health. It is presumed that several pregnancy complications such as PE, preterm birth, gestational diabetes could be the effect of Vit D deficiency as well as complications manifesting in offspring later in life such as asthma, psychomotor development, and cognitive disorders [15,16,17].

The paper aims to present the possibility of Vit D use in preventing PE, emphasizing its impact on the pathogenetic elements of preeclampsia development.

## 2. Pathogenesis of Preeclampsia 

Despite significant advances in research on PE pathophysiology, its cause has not been definitively settled. It has been demonstrated that its development is associated with the presence of the placenta, and the processes that initiate it begin at the time of abnormal trophoblast invasion in early pregnancy. As a result, they lead to the development of trophoblast/placental hypoxia and consequently to the development of oxidative stress and endothelial dysfunction in the later phases of the disease, which are manifested by clinical symptoms. The only effective way to treat PE is delivery which indicates its relationship with the presence of the placenta.

A two-stage model for PE development has been proposed. The first stage involves incomplete remodeling of spirals arteries in the uterus, which leads to hypoxia of the placenta. In the second stage, anti-angiogenic factors responsible for endothelial damage are released from the hypoxic placenta into the maternal circulation.

The trophoblast implantation involves its invasion into the uteroplacental arteries and then their transformation into dilated, inelastic tubes, which provides increased blood flow without maternal vasomotor control. The purpose of this process is to provide increased perfusion of the intervillous space. In the case of inadequate trophoblast invasion and lack of transformation of spiral arteries, relative hypoxia of the placenta with the development of oxidative stress occurs [18]. Trophoblast hypoxia could explain the death of cells, mainly in the mechanism of apoptosis [19,20]. These processes occur early in pregnancy; trophoblast implantation is completed by the 16–17th week. The critical issue remains the cause of abnormal trophoblast implantation. Many researchers suggest an impaired response of the maternal immune system or abnormal development of maternal immune tolerance to the development of the allogenic fetus [20,21].

Several studies have been conducted on immune changes within the preeclamptic decidua. They have shown excessive activation of neutrophils and monocytes, which synthesize large amounts of pro-inflammatory cytokines such as IL-1β, IL-6, and IL-8 [22,23].

In addition, CD4+ and CD8+ T cells together with natural killer cells (NKc) and dendritic cells (DCs) show a different response in women with PE compared to healthy pregnant women [24,25]. An animal model has shown that decidual natural killer cells (dNKc) knockout mice did not develop spiral arteries [26]. It has been revealed that dNKc, by releasing pro-apoptotic factors during normal pregnancy, can lead to apoptosis in vascular smooth muscle cells (VSMC) and endothelial cells, which are essential in the process of spiral arteries remodeling [27].

Abnormal remodeling of spiral arteries entails a disorder of placental function, which is the source of many factors entering the maternal circulation responsible for increased inflammatory response, oxidative stress, apoptosis, and generalized endothelial dysfunction, which is an essential pathophysiological change in PE, explaining the development of clinical symptoms [28]. These include generalized vasoconstriction and restricted organ perfusion. Factors that adversely affect endothelial function such as obesity, diabetes, malnutrition intensify the maternal response to signals from the hypoxic placenta and thus promote PE development [29]. It has been suggested that endothelial dysfunction could be more pronounced in PE than in GH, which explains less severe clinical symptoms and a better prognosis [30,31].

The endothelium has autocrine, paracrine, and endocrine properties. It is responsible for the synthesis of numerous vasodilators (nitric oxide (NO), prostacyclin I2 (PGI2), endothelium-derived hyperpolarizing factor (EDHF), bradykinin, histamine, serotonin, substance P), and vasoconstrictors (endothelin-1 (ET-1), angiotensin II (ANG-II), thromboxane A2 (TX2), prostacyclin H2 and reactive oxygen species (ROS)). The imbalance between them and the predominance of the synthesis of vasoconstrictive factors are responsible for developing many pathological processes, including preeclampsia. Endothelial dysfunction is connected with the presence of at least one of the following changes: the decrease in the NO synthesis and bioavailability, higher adhesion molecules and inflammatory genes expression, intensified ROS synthesis, impaired endothelium-dependent vasorelaxation, decreased fibrinolysis and enhanced endothelial permeability [32]. Hypoxia and oxidative stress have been thought to disrupt the placental synthesis of pro-angiogenic and anti-angiogenic factors, which play a key role in the pathogenesis of PE [33]. It is characterized by a reduced concentration of pro-angiogenic factors and a predominance of anti-angiogenic factors [34]. The characteristic shift in balance favoring anti-angiogenic factors is present from the beginning of pregnancy and impairs the trophoblast implantation [35,36].

The essential pro-angiogenic factors in pregnancy are vascular endothelial growth (VEGF) and placental growth (PlGF). VEGF plays an important role by attaching and activating the two-cell surface receptor tyrosine kinases, vascular endothelial growth factor receptor-1 (VEGFR-1/Flt-1). Furthermore, vascular endothelial growth factor receptor 2/kinase insert domain receptor (VEGFR-2/KDR), which is present on endothelial cells, stimulates their proliferation and the release of the plasminogen activators [37]. Its pro-angiogenic activity is expressed through these mechanisms [38]. VEGF has been postulated to play an important role in maintaining endothelial integrity. A link between VEGF and placental oxidative stress has been suggested. In patients with severe preeclampsia, changes in VEGF concentration resulting from hypoxia may cause an increase in the activity of 5’ adenosine monophosphate-activated protein kinase (AMPK) [39]. AMPK plays an important role in many of the cellular energy and metabolic processes. It affects angiogenesis within the placenta, and its activity increases under hypoxia conditions observed in preeclampsia [40].

Another pro-angiogenic factor important for the proper development of pregnancy is PlGF which regulates endothelial cell adhesion and chemotaxis. PlGF is thought to enhance the pro-angiogenic effect of VEGF [41,42]. The transforming growth factor-β (TGF-β) family has been shown to play an important role in endothelial cell growth and angiogenesis, modulates the immune response and thus regulates many placental functions [43]. It has been found that TGF-β enhances the expression of VEGF, and its concentration is significantly reduced in PE [44]. The main anti-angiogenic agents whose role in the pathogenesis of preeclampsia has been described are VEGF receptors (VEGFR1 and VEGFR2) and soluble endoglin (sEng). VEGFR1 is also known as fms-like tyrosine kinase-1 (sFlt-1) [45]. It has been shown that sFlt-1 by binding VEGF and PlGF reduces the formation of vessels within the trophoblast [46,47]. It has been observed that an increase in its levels accompanied by a decrease in PlGF concentration correlates with the PE severity [48].

With the limited perfusion and hypoxia that characterize PE, the placenta produces large amounts of sFlt-1 and sEng, one of the potent anti-angiogenic factors, which both are thought to be responsible for endothelial damage and PE symptoms [49,50]. It has been shown that sEng by disturbing TGF-β1 signaling in endothelium cells reduces vasodilation and limits the pro-angiogenic effect [51]. On the pregnant rodents model, Venkatesha et al. have shown that the administration of sEng significantly increases blood pressure and develops mild proteinuria. In contrast, the administration of sFlt-1 results in the development of severe hypertension and severe proteinuria and the appearance of HELLP (hemolysis, elevated liver enzymes, low platelets count) syndrome symptoms. sENG together with sFlt-1 can inhibit the action of both TGF-β1 and VEGF [52].

These observations confirm the results of studies by other authors recognizing sFlt-1 as the main anti-angiogenic factor involved in the PE development [46].

It has been reported that the activation of eNOS (endothelial nitric oxide synthase) and the NO release, the potent vasodilator, is inhibited by sEng, which significantly limits the proper growth and invasion of the trophoblast [53]. On the other hand, VEGF and PIGF positively affect the synthesis and bioavailability of NO [54,55].

sFlt-1 by inhibiting PlGF and VEGF leads to a decrease in NO synthesis, which is additionally disturbed by oxidative stress and ROS. These observations confirm that the synthesis and release of NO are dependent on the balance between pro-angiogenic and anti-angiogenic factors. Disturbance of this balance in favor of anti-angiogenic factors adversely affects the release of NO [56].

Increased inflammation observed in PE, which is expressed for example by elevated TNF-α (tumor necrosis factor α) concentrations, is associated with an increase in the expression of adhesive molecules ICAM1 (intercellular adhesion molecule 1), VICAM1 (vascular cell adhesion molecule 1), and endothelin 1 (ET-1), the potent vasoconstrictor, which are all markers of endothelial damage [57,58,59].

The mechanism of action of anti-angiogenic factors and the imbalance between pro- and anti-angiogenic factors partly explain the stages of the pathogenetic pathway in the development of PE. In addition, the assessment of the sFlt/PIGF ratio is of prognostic importance to predict the severity of PE complications: the increased sFlt/PlGF ratio anticipates the appearance of adverse outcomes within two weeks [60,61].

Hypoxia-inducible factor α (HIF1α) is a molecular factor that combines placental hypoxia with downstream mediators of PE. The synthesis of HIF1α has been shown to be intensified in placental hypoxia. It has also been observed that HIF1α is a factor inducing the synthesis and release of sFLT-1 in placental explants [62].

During a healthy pregnancy, there is an increase in metalloproteinases (MMs) activity to ensure proper trophoblast implantation which requires the destruction of the extracellular matrix. The invasive potential of extravillous trophoblast (EVT) cells relates to MMP-2 and MMP-9 expression [63]. Reduced activity of metalloproteinases is associated with PE development [64]. This observation is confirmed by the results of studies indicating the relationship of vasoconstriction typical for PE with reduced expression of MMP-2 and MMP-9. Chen et al. have reported a different effect of pro- and anti-angiogenic factors on MMP-2 activity in placental tissues and vascular wall. sFlt-1 lowered the activity of these molecules, and VEGF reversed this process and improved placentation [65].

During physiological pregnancy, the phenomenon of increased production of PGI2 as a platelet inhibitor and vasodilator and a limitation of the synthesis of TX2 responsible for platelet activation and vasoconstriction is observed. In PE, endothelial dysfunction results in the peroxidation of endothelial lipids and the limitation of antioxidant processes. Lipid peroxidation activates cyclooxygenase (COX—cyclooxygenase), which is responsible for the synthesis of TX2 thromboxane, disturbing the TX2/PGI2 balance in favor of TX2 [66]. Although progesterone is the hormone responsible for the proper development of pregnancy, its excess can lead to a decrease in the synthesis of prostacyclin and an increase in the production of thromboxane [67].

In a healthy pregnancy, activation of the renin-angiotensin-aldosterone system (RAAS) is observed, which leads to an increase in the concentration of renin, angiotensinogen, and angiotensin II [68]. Many authors have so far postulated that RAAS has a significant impact on the development of preeclampsia. In PE, RAAS is inhibited, confirmed by a reduced serum concentration of angiotensin I, angiotensin II, aldosterone, an increase in renin plasma activity, and the concentration of antibodies to the angiotensin II type 1 receptor (ATR1-AA). These antibodies are responsible for stimulating the signaling ATR1 and, as a result, for increasing blood pressure [69,70,71]. However, it seems that the role of this system in the pathogenesis of preeclampsia has not been definitively determined. Many researchers believe that it has a significant impact on the development of PE. However, there is a growing body of evidence that although RAAS plays an important role in the development of pregnancy, its importance in the pathogenesis of PE is not significant except ATR1-AA [72].

Current reports also emphasize the importance of disorders of the methionine-homocysteine system and cellular mechanisms of oxygen sensing in the process of abnormal trophoblast invasion and placental hypoxia [73,74]. Hyperhomocysteinemia is associated with PE development, and it is thought to be responsible for endothelium dysfunction caused by thrombosis [75]. One of the causes of hyperhomocysteinemia is MTHFR 677TT genotype, whose relationship with the PE development is postulated. Micronutrients such as folic acid and riboflavin have been shown to reduce homocysteine levels significantly [76].

Figure 1 shows the main stages in PE pathogenesis.

## 3. Vitamin D

### 3.1. Metabolism 

For many years, vitamin D has been classified along with other vitamins. It is currently known to play the role of a prohormone—a precursor of the steroid hormone calcitriol (1,25-dihydroxyvitamin D-1,25(OH)2D). Vitamin D is a group of fat-soluble sterols, the most important of which are vitamin D2—ergocalciferol and vitamin D3—cholecalciferol. Plants and fungi synthesize vitamin D2, and animals produce vitamin D3. The most important source of vitamin D for humans is its production in the skin. The synthesis of vitamin D3 is influenced by UVB intensity and skin pigmentation [77]. UVB intensity varies depending on the season and latitude. Melanin contained in the skin blocks the synthesis of vitamin D3 like sunscreens or clothing [78]. Food and supplements are also the sources of vitamin D, although its dosage remains a matter of dispute. The sources of vitamin D include fatty fish, fish liver oil, egg yolk. In fatty fish, vitamin D3 is present, while vitamin D2 is in other enriched products. In most foods, except fortified products, the vitamin D content is low [79].

Vitamin D3 is synthesized from the liver-derived precursor 7-dehydro-cholesterol (7-DHC) under the influence of ultraviolet B radiation (UVB) with a wavelength of 290–315 nm through a membrane-enhanced thermal-dependent isomerization reaction. Vitamin D3 enters the circulation through the capillaries and preferentially reversibly binds in vast majority to vitamin D binding protein (VDBP)—a plasma glycoprotein produced by the liver and, to a lesser extent, to plasma albumin [80].

As vitamin D3 and D2 share the same metabolism and exert the function of prohormone, the presented review does not make (unless it is indicated in the text) a distinction between vitamin D2 and D3 and the term vitamin D (Vit D) is used for both these forms.

The first stage of transforming vitamin D2 and D3 is 25-hydroxylation, which occurs mainly in the liver with the participation of the mitochondrial form of 25-hydroxylase (CYP27A1), which appears to be a bifunctional cytochrome P450 enzyme [81]. At this stage, 25-hydroxyvitamin D—calcidiol (25(OH)D), the main form of circulating Vit D is synthesized. Its synthesis is also possible in other tissues in the auto- and paracrine way. The measurement of serum 25(OH)D concentrations allows assessing the status of Vit D because it best reflects the supply of Vit D from all sources [82]. Calcidiol levels are influenced by several factors such as population factor, latitude, sun exposure, air pollution, gender, eating habits, or religious recommendations (clothing in Muslim women), which may be responsible for up to 50% in variations in serum 25(OH)D levels.

Free 25(OH)D accounts for less than 1% of the total 25(OH)D, while the bioavailability fraction, which consists of free 25(OH)D and bound to albumin, is up to 15% of the total 25(OH)D [83]. Some authors believe that only free 25(OH)D is responsible for biological effects; hence, assessing its concentrations would be more reliable concerning its activity. It has been shown that the concentrations of 25(OH)D reflect satisfyingly the level of free 25(OH)D, but in states with significant changes in the concentrations of binding proteins such as pregnancy, the total 25(OH)D level does not fully reflect its bioavailability [84].

25(OH)D binds to VDBP, and this complex is subject to endocytic internalization in the kidney proximal tubule cells in which the transmembrane protein megalin participates [85]. Megalin-mediated endocytosis of 25(OH)D/VDBP also requires the receptor-associated protein (RAP) and cubilin, a protein necessary for sequestering VDBP on the cell surface before internalization by megalin. This mechanism is the most critical element allowing the hydroxylation of Vit D in kidney cells [86].

The next stage of Vit D activation occurs in the kidneys under the influence of the cytochrome P450 enzyme—25(OH)D-1α-hydroxylase (CYP27B1) [87]. It leads to the synthesis of the active form of vitamin D—1,25-dihydroyxvitamin D (1,25(OH)2D)—calcitriol which also binds to VDBP, and this form is found in plasma [88]. The affinity of 25(OH)D and 1,25(OH)2D is significantly higher to VDBP than to albumin [89]. VDBP is a multifunctional protein that, in addition to transporting Vit D, has several other functions. The most important is the modulation of the inflammatory response and the control of bone development. VDBP alleles and variants have been shown to be associated with different susceptibility to diseases, including autoimmune diseases [90].

Bone and mineral metabolism control the synthesis of 1,25(OH)2D. To maintain adequate calcium concentrations in the kidneys, intestines, and bones, parathyroid hormone (PTH) stimulates the formation of 1,25(OH)2D through a mechanism dependent on cAMP (cyclic adenosine monophosphate), while fibroblast growth factor 23 (FGF-23) inhibits its synthesis. CYP27B1 activity is also controlled by calcitriol alone by the negative feedback regulatory loop [91]. The synthesis and release of PTH are suppressed by calcitriol and FGF23, which block CYP27B1 activity in response to elevated phosphate levels [92]. In this way, calcitriol inhibits its synthesis by restraining the synthesis of PTH, direct transcriptional repression of the CYP27B1 gene, and by activating FGF23 and 1,25-(OH)2D-24-hydroxylase (CYP24A1), which is the enzyme responsible for the conversion of calcitriol to biologically less-active metabolites [93].

Vitamin D2 is produced by fungi and plants under the influence of solar radiation and is structurally different from vitamin D3, which explains the lower affinity of vitamin D2 to VDBP. It results in faster clearance from the circulation and limited conversion to 25(OH)D [94]. Hence, even daily supplementation with Vit D2 is not as effective in preventing deficiency as the supply of vitamin D3 [95].

The activity of 25-hydroxyvitamin D-1α-hydroxylase (1α-hydroxylase) has been demonstrated in many other tissues and organs besides the kidneys, including the circulatory system. It has been accredited that the local synthesis of 1,25(OH)2D is primarily dependent on the availability of 25(OH)D, but there are many arguments for the fact that its synthesis is also influenced by other factors such as cytokines [91].

The biological effects of 1,25(OH)2D may be genomic and non-genomic mediated. All genomic effects of 1,25(OH)2D are VDR (vitamin D receptor) mediated. VDR shows high affinity and specificity to 1,25(OH)2D [96].

After activating the cytosolic VDR by ligand binding and transfer to the cell nucleus, the activated VDR interacts with vitamin-D response elements (VDRE) in the promoter region of vitamin D target genes and is responsible for stimulating co-activators or co-repressors to induce or repress basal transcription processes that thus regulate gene transcription [97]. In this way, the activity of about 3% of the genome is regulated by vitamin D [98]. VDRs are present in many cardiovascular system cells, such as endothelial cells, cardiomyocytes, blood vessels, smooth muscle cells, most immune cells, and platelets [99,100,101]. Active VDR has an inhibitory effect on several genes, including the gene responsible for the synthesis of PTH and CYP27B1 [99]. The primary function of Vit D, in which VDR participates, is to control calcium and phosphorus metabolism [102].

It has also been demonstrated that Vit D is responsible for the expression of many genes influencing activation, proliferation, and differentiation of several cells, including cells of the immune system. This explains its potential impact on the pathogenesis of many cardiovascular and autoimmune diseases, infections, and cancer [103,104].

VDR mutation, a rare autosomal recessive disorder, is responsible for the hereditary Vit D-resistant rickets (HVDRR), which is characterized by a lack of response to vitamin D [105].

The action of calcitriol on the non-genomic pathway within cells takes place through second messengers generated by membrane-initiated signaling pathways. VDR and the membrane-associated rapid response steroid-binding protein (MARRS), which are present in the cell membrane after the attachment of calcitriol, activate protein pathways such as kinase C (PKC) mitogen-activated protein kinase (MAPK), protein kinase A (PKA), phosphatidylinositol phosphate, and Ca^2+^ and chloride channels [106,107].

The activation of such several second messengers explains the diversity of biological effects of Vit D, which include not only the effects on bones and mineral balance known for many years but also the impact on the cell cycle (growth, division, apoptosis), the immunomodulatory effect, and activation of cathelicidins [108].

Results from previous studies have suggested the expression of CYP27B11 in non-renal cells which explains the local synthesis of 1,25(OH)2D3 and its effect on the transcriptional regulation of genes. The activity of the extrarenal form of CYP27B11 is regulated in a tissue-specific manner. Within immune cells (monocytes, macrophages), CYP27B1 is activated by pro-inflammatory cytokines such as interferon-γ (INF-γ) and TNF-α, and the concentration of calcitriol does not affect its activity [109,110].

It has been suggested that the most important factor regulating the extrarenal synthesis of calcitriol is the availability of 25(OH)D [111].

### 3.2. Mechanism of Action 

Vitamin D 3 deficiency is common worldwide, and even in countries with appropriate insolation, its incidence is high and estimated at 30–50% [112,113]. It may result from the insufficient synthesis in the skin or deficiencies in the diet. The first cause is common in people with low sun exposure (living in the north, spending much time indoors, the elderly, dark-skinned, wearing covering clothes or using sun blockers). In women and children in northern European countries, an increase in the prevalence of vitamin D deficiency is observed due to concern for the development of skin cancer and the widespread use of sunscreens [114]. Vitamin D deficiency caused by diet is the effect of a type of diet (vegan) or malabsorption from the gastrointestinal tract resulting from various diseases (e.g., coeliac disease, pancreatic insufficiency, cystic fibrosis) [16]. Impaired kidney and liver function are also responsible for decreased Vit D levels [115]. Vitamin D catabolism is accelerated by some drugs such as glucocorticosteroids, calcium channel blockers, and anticonvulsants, leading to its deficit [116].

The degree of vitamin D deficiency in the general population was presented by the Australian and New Zealand Bone and Mineral Society and Osteoporosis (Table 1) [117].

According to the Institute of Medicine (IOM) recommendations, the daily dose of Vit D is 600 IU/day for people aged 1–70 years and 800 IU/day for people over 70 years of age. Such doses should provide a level of 25(OH)D of at least 50 nmol/L [118]. Vitamin D toxic effects can manifest with its long-term use in doses above 4000 IU/day and at a 50–150 ng/mL concentration. Acute symptoms of intoxication are usually caused by Vit D intake above 10,000 IU/day with expected concentrations of 25(OH)D > 150ng/mL [119]. The Endocrine Society has proposed to include Vit D levels in the risk of intoxication assessment. Concentrations of 25(OH)D > 100 ng/mL (250 nmol/L) was defined as hypervitaminosis, while a concentration of > 150 ng/mL (375 nmol/L) was defined as intoxication [120]. Vitamin D intoxication is characterized by hypercalcemia and the associated symptoms. These include manifestations from the central nervous system (confusion, psychosis, stupor, or coma), renal (hypercalciuria, acute kidney injury, dehydration, and nephrocalcinosis), gastrointestinal (abdominal pain, vomiting, polydipsia, anorexia, constipation, pancreatitis), and cardiovascular circulatory system (hypertension, shortened QT interval, ST-segment elevation, bradyarrhythmias, first-degree heart block) [121].

#### 3.2.1. Calcium and Phosphorus Metabolism

The complex of 1,25(OH)2D and VDR as transcription factor enhances the expression of the gene encoding calcium-binding protein, which intensifies the absorption of calcium and phosphorus. Calcitriol is also involved in bone metabolism: formation, resorption, mineralization, and the maintenance of neuromuscular function. Vitamin D, by influencing the increase in calcium concentrations and the mechanism of negative feedback, inhibits the synthesis of parathyroid hormone [122].

#### 3.2.2. Immune System 

1,25(OH)2D exerts anti-inflammatory activity demonstrated in several experimental models through multiple mechanisms. The importance and participation of 1,25(OH)2D in immune processes is confirmed by the presence of CYP27B1 in immune cells such as inactivated CD4+ and CD8+ T cells, B cells, macrophages, and DCs and their ability to local calcitriol synthesis [123,124].

The Vit D influence on the immune system involves an innate and adaptive response. The innate immune response is realized by activating Toll-like receptors (TLRs), which are present on many immune system cells and endothelial cells. Activation of TLRs results in synthesizing antimicrobial peptides (AMPs) such as cathelicidin and ROS, whose prominent role is to fight the pathogenic microorganism. 1,25(OH)2D has been observed to enhance cathelicidin expression in endothelial cells and myeloid cells [125]. The adaptive response involves cells presenting the antigen, mainly DCs and macrophages and cells recognizing the antigen: T and B lymphocytes. It has been observed that Vit D may inhibit the adaptive immune system. The maturation of DCs and the ability to present antigen and activate T cells are significantly limited by 25(OH)2D [126].

It is supposed that Vit D is responsible for increasing the number of regulatory T cells [127]. In activated leukocytes, NF-κB (nuclear factor kappa B) expression is directly inhibited by the 1,25(OH)2D-VDR complex [128], which results in a reduction in the synthesis of the pro-inflammatory cytokines such as IL-1, IL-2, IL-6, IL-8, IL-23, TNF-α, and INF-γ (Interferon-γ) [129]. Additionally, it has been reported that 1,25(OH)2D intensifies the release of anti-inflammatory cytokines such as IL-4 and IL-10 [130]. The results presented by Noyola-Martinez et al. have suggested that cytokines regulate calcitriol metabolism in the human placenta; specifically, INF-γ may contribute to calcitriol production while TNF-α favors its catabolism [131].

It has been shown that the reduction in TNF-α and IL-6 synthesis by 1,25(OH)2D and 25(OH)D is the result of their effect on monocyte/macrophage mitogen-activated protein kinase phosphatase-1 [132]. Wu et al. have found that 1,25(OH)2D downregulates pro-inflammatory microRNA-155 production in macrophages, leading to the stimulation of the suppressor of cytokine signaling 1. An effect of 1,25(OH)2D on the reduction of Toll-like receptor-mediated inflammatory response has also been observed [133].

Despite the beneficial Vit D effect on reducing inflammation demonstrated in many observational and experimental studies, the results of randomized controlled trials (RCTs) are inconclusive. The selected inflammatory parameters such as TNF-a and C reactive protein concentrations have been shown to be reduced after Vit D supplementation, but some studies have not confirmed this effect [134,135,136].

It is thought that the regulation of adaptive response by 1,25(OH)2D and its analogues reduce the risk of developing autoimmune diseases such as lupus and rheumatoid arthritis, diabetes mellitus type 1, and multiple sclerosis [137], and they may serve as adjuncts to immunosuppressants following transplantation procedures [138].

The anti-inflammatory effect of 1,25(OH)2D is also expressed by inhibiting the synthesis of prostaglandins (PGs) [139]. Studies conducted on cancer cell lines have shown that under the influence of 1,25(OH)2D, the expression of cyclooxygenase (COX-2), responsible for the synthesis of PGs, is inhibited. In addition, the PGs-degrading enzyme 15-hydroxyprostaglandin dehydrogenase activity is intensified by 1,25(OH)2D [140].

Calcitriol has protective effect on normal cells by limiting apoptosis caused by many factors [141,142,143,144].

#### 3.2.3. Cardiovascular System 

Vitamin D deficiency is now recognized as one of the risk factors for developing cardiovascular diseases [145,146]. It has been shown that Vit D deficiency through increased vascular resistance and vasoconstriction leads to hypertension [147].

It has also been found as one of the risk factors for death in course of cardiovascular diseases (CVD) and death from cancer [145,148]. Many studies have shown an increased risk of heart attack, stroke, circulatory failure, and peripheral vascular disease with limited sun exposure and thus lower serum concentrations of 25(OH)D [149,150].

It has been observed that the incidence of the above complications and mortality from CVD increases during the winter months when Vit D levels are at their lowest. It suggests its protective role in CVD prevention [151]. It has been shown that the potential role of Vit D in CVD prophylaxis consists in a favorable effect on the vascular endothelium, regulation of blood vessels tone and blood pressure [99,150,152]. Vitamin D also reduces inflammatory processes, insulin resistance, improves fat metabolism, and reduces the calcification of blood vessels [150,153,154].

Studies evaluating the relationship of Vit D concentration with CVD have shown that in the case of low Vit D concentration < 15 ng/mL, the risk of developing hypertension is three times higher than levels > 30 ng/mL [155]. Such a low Vit D concentration was also associated with a 60% risk of other cardiovascular diseases [149]. Similar conclusions have been presented by Giovannucci et al., who observed a two-fold increase in the rate of heart attacks at low Vit D concentrations compared to level > 30 ng/mL [156]. Based on the conducted studies, it appears that the 25(OH)D level of about 36 ng/mL is the concentration above which no further reduction in mortality or cardiovascular complications is observed [157]. Similarly, the incidence of hypertension has been found to be inversely correlated to Vit D levels [155]. It has been postulated that a high concentration of 25(OH)D may be responsible for an increase in the concentration of adiponectin, which has a protective effect on the cardiovascular system [158]. Unfortunately, the results of observational studies were not confirmed by meta-analyses, which assessed the effectiveness of Vit D supplementation in preventing CVD [150,159]. Differences in the results of observational studies and RCTs are likely due to the failure to consider the Vit D concentration in study participants before starting its supplementation and different doses and administration regimens used [160]. This is confirmed by the observations of Amrein et al. who have shown that among patients in an intensive care unit, Vit D supplementation reduced the risk of death only in those with a concentration of 25(OH)D < 30 nmol/L (12 ng/mL) [161].

Hence, many researchers recognize that Vit D concentration represents general health. It is also confirmed by the fact that its deficiency is associated not only with diseases of the cardiovascular system, but also with inflammation, glucose metabolism disorders, weight gain, infectious diseases, multiple sclerosis, mood disorders, declining cognitive function, impaired physical functioning, and all-cause mortality [162].

The results of many studies have indicated that Vit D has a significant, beneficial effect on the endothelium and angiogenesis. The enzyme CYP27B1 is present and active in endothelial cells, suggesting local synthesis of 1,25(OH)2D. Its presence has been demonstrated in isolated human umbilical vein endothelial cells (HUVECs) and endothelial cells from human renal arteries, postcapillary venules from lymphoid tissue [163]. According to many researchers, the effect of 1,25(OH)2D varies significantly depending on the type of tissue. Studies conducted on healthy endothelial cells have suggested that within them, it expresses a pro-angiogenic effect. In contrast, in endothelial cells derived from cancerous tumors, it behaves as an anti-angiogenic factor. The pro-angiogenic effect of 1,25(OH)2D results in the intensification of VEGF expression and pro-MMP2 activation. It leads to the enhanced formation of capillary-like structures and cell proliferation of endothelial colony-forming cells (ECFCs). This effect has been demonstrated within HUVECs [164]. Similarly, the increased VEGF expression and blood vessel formation were demonstrated in an animal model under the influence of 1,25(OH)2D analogue [165]. It has been observed that 1,25(OH)2D enhances VEGF expression and has a positive effect on vascularization [166].

On the other hand, there are reports of adverse effects of 1,25(OH)2D on the endothelium and its anti-angiogenic effects. Interestingly, they concern observations made in cancerous tissues. Mantell et al. have shown that 1,25(OH2D inhibits VEGF- and VEGF-dependent cell sprouting, elongation, and proliferation processes. In addition, it is supposed to be responsible for sprouting endothelial cells in vitro. Based on the results of their research, these authors have concluded that 1,25(OH)2D could be useful in the prevention and treatment of conditions with pathological angiogenesis [167].

1,25(OH)2D also affects other mechanisms that regulate endothelial function and angiogenesis. The 1,25(OH)2D-VDR complex has been shown to enhance the activity of cystathionine b-synthase, the enzyme responsible for the metabolic elimination of homocysteine. It has been revealed that vitamin D level < 30 ng/mL is an independent risk factor for hyperhomocysteinemia (RR 1.89, 95% CI 1.41–2.52) [168]. Hyperhomocysteinemia is postulated to be one of the important risk factors for developing CVD and neurogenerative diseases. The observed protective effect of Vit D supplementation on endothelial cells consists in reducing oxidative stress by increasing glutathione levels and limiting lipid peroxidation [169,170].

In the light of the results of several studies, the beneficial impact of vitamin D on the cardiovascular system is based on its inhibitory effect on RAAS [171,172,173,174]. The mechanism of favorable action of vitamin D consists in suppressing the renin gene transcription by blocking the activity of the cAMP response element in the promoter of the renin gene, for which active VDR is responsible [175]. It has also been reported that activated VDR by limiting the expression of the angiotensin II receptor reduces the synthesis of ROS and thus improves the endothelial condition [176].

Another mechanism explaining the beneficial effect of Vit D on hypertension is the inhibition of PTH synthesis [98,177]. It has been observed that PTH intensifies oxidative stress, which is responsible for endothelial dysfunction and aldosterone release, and intracellular calcium overloading of cardiomyocytes with myocardium hypertrophy [178,179]. On the other hand, however, in people with VDR mutation, which leads to the development of resistance to 1,25(OH)2D, no significant differences in RAAS activity, blood pressure values or echocardiographic results were observed compared to healthy people [180].

## 4. Vitamin D in Pregnancy 

Calcitriol and VDR are present within the tissues of the female reproductive organ, and their presence has been shown in the uterus, fallopian tube, ovary, placenta, and the pituitary, hypothalamus, and mammary glands. Pregnancy significantly affects the metabolism of Vit D. Its concentration increases 2–3 times in the first weeks of pregnancy, and maternal kidneys are its primary source [181,182]. However, there are also reports indicating that its levels remain constant during pregnancy, similar to those of non-pregnant women [183] or even decrease [184]. The placenta not only enables the transfer of 25(OH)D from the mother to the fetus but is also the place of its production, which is confirmed by the presence of CYP27B1 in the decidua and placenta [109].

The postulated progressive increase in the concentration of 1,25(OH)2D in the early first trimester results from enhanced CYP27B1 activity in the maternal kidneys and the activation of an additional source of its synthesis within trophoblast and decidua [185]. This rise is observed in pregnancy before the fetus manifests an increased need for calcium [184].

The activity of CYP27B1 in the kidneys during pregnancy is similar to its extrarenal activity. A relative lack of sensitivity to the high concentration of PTH has been observed. In pregnancy, PTH does not inhibit calcitriol synthesis, just as it is not inhibited by a high level of 1,25(OH)2D. Hence, it seems that the increase in 1,25(OH)2D in pregnancy is independent of the PTH levels [186]. The CYP27B1 activity is controlled by hormones related to pregnancy, including estradiol, prolactin, and placental lactogen [184].

1,25(OH)2D, when combined with VDR, stimulates organogenesis in a genomic and nongenomic-mediated manner. Genes containing VDRE are responsible for several processes essential for the development of pregnancy, such as bone and mineral metabolism, cell life and death (comprising proliferation, differentiation and apoptosis), and immune function—both innate and adaptive immunity [187].

The role of Vit D in the controlling of immune processes in the maternal-fetal unit is considered by some authors to be crucial for proper pregnancy development [188].

The increase in VDBP concentrations affects the availability of a functionally active, accessible form of Vit D. In pregnancy, it reflects better the status of vitamin D than a total 25(OH)D. It is thought that in pregnancy it reflects better the status of vitamin D than a total 25(OH)D. Indeed, despite the increase in the concentration of VDBP and total 25(OH)D, it has been shown that between the 15th–36th week the levels of the free biologically active form 25(OH)D are significantly reduced [189]. Hence, even normal 25(OH)D levels may not reflect the actual 25(OH)D bioavailability, and in some pregnant women, the Vit D deficiency could be challenging to recognize [190,191].

There is a growing body of evidence that VDBP could be a good predictor of the development of pregnancy complications such as PE, gestational diabetes mellitus (GDM) or preterm birth, although the literature on this issue is so far scarce [192,193].

In the early pregnancy, characterized by intensive cell division and growth, maternal 25(OH)D and fetal 1,25(OH)2D produced from about the 6–10th week can interact with different signaling systems to control organ development. VDR expression in many fetal and trophoblastic cell types has been confirmed [194].

Studies in an animal model have shown that Vit D deficiency affects the inhibition or intensification of the expression of specific genes within the placenta [195].

It is thought that Vit D has a beneficial effect on the development and function of the trophoblast by regulating calcium transport and modulating immune processes [196]. It has been observed that its deficiency affected changes in T-cell phenotype, which is associated with preterm birth [197]. A link between vitamin D deficiency and recurrent miscarriages has also been reported [198].

There is a lot of available data to support the observation that adequate levels of Vit D increase the fertilization ratio, and it is dependent on BMI (body mass index). It has shown to decrease proportionally to the increase in BMI (4% decrease per 1 kg/m^2^ weight gain, with a BMI above 29 kg/m^2^). The reduced concentration of Vit D bioavailability in women with obesity through inadequate modulation of immune processes at the fetal-maternal interface may partly explain the higher incidence of pregnancy complications in this group of women. Observations have indicated that this is a group of women significantly exposed to Vit D deficiency due to reduced exposure to sunlight and a high-calorie diet with a low vitamin D content [199].

In animal and human models, it has been shown that 25(OH)D easily crosses the placenta and its levels are 70–100% of the maternal concentration in umbilical blood [181,200]. It has been reported that the synthesis of 1,25(OH)2D in the fetus may be limited by high levels of calcium, phosphorus, and low concentrations of PTH demonstrated in the umbilical blood [201].

### Recommendations on Supplementation of Vitamin D 

Vitamin D deficiency is commonly found among pregnant women in various ethnic populations [202,203]. Clothing with minimal skin exposure, increased urbanization, skin pigmentation, and vegetarian diets all are supposed to have contributed to Vit D deficiency epidemic worldwide [204,205]. In women with lower levels Vit D, i.e., in dark-skinned and Muslim women, there is a higher incidence of preeclampsia [206,207,208].

Vitamin D deficiency is also observed in breastfeeding women without supplementation [16].

Currently, there is no clear definition of Vit D deficiency in pregnancy based on the assessment of 25(OH)D concentrations as well as one prophylactic recommended dose of vitamin D (Table 2) [209,210].

Since obesity adversely affects the status of Vit D, its higher dose is proposed for pregnant women with obesity. There were no differences in the synthesis of vitamin D in the skin in women with obesity and women with average body weight, while it is estimated that the level of vitamin D in obesity is lower by as much as 57%. This phenomenon is explained by releasing vitamin D into the circulation hindered by excessive adipose tissue [211].

Currently, no organization proposes the use of vitamin D in PE prevention. So far, there is no data in the literature on whether vitamin D supplementation commonly recommended in many countries affects the incidence of preeclampsia.

It has been shown that Vit D deficiency increases during pregnancy, which is the result of an increase in demand. Statistics suggest that it may affect up to 50–100% of all pregnant women if the lower limit of the norm is the level of 50 nmol/L and 15–84% if the lower limit is 25 nmol/L [212]. According to Hollis et al. only the level of 25(OH)D of about 100 nmol/L (40 ng/mL) ensures the synthesis of 1,25(OH)2D appropriate for normal pregnancy development [213]. McDonnell et al. have shown that maternal 25(OH)D concentrations ≥ 100 nmol/L reduce the risk of preterm birth by 59% compared to <50 nmol/L [214].

Results of research conducted by Mumford et al. have revealed that Vit D levels above 75 nmol/L are associated with a higher probability of conception, a reduced risk of pregnancy loss, and a higher rate of live birth [215]. There are opinions that this level of 25(OH)D should be achieved at the beginning of pregnancy because only then its modulating effect on immunity function, necessary for the proper development of the fetus and the course of pregnancy, could be significant [216,217].

Unfortunately, despite the potential beneficial effect of Vit D on several pathophysiological processes leading to the development of pregnancy complications and promising results of observational studies, most RCTs and meta-analyses have not demonstrated in a definite and unambiguous way the effect of Vit D supplementation on reducing the incidence of pregnancy complications, including preeclampsia [218,219,220].

The lack of clear criteria for diagnosing Vit D deficiency in pregnancy seems to be the main problem. At the same time, there are no recommendations to determine its level in the periconception period or the first trimester, even in patients at risk of developing pregnancy complications [209].

Regardless of the potential benefits of vitamin D supplementation for the pregnant woman, it should be emphasized that the fetus is fully dependent on maternal 25(OH)D.

It has been widely accepted that the minimum level of 25(OH)D in cord blood, which ensures the proper development of the skeletal system in the fetus, is >25–30 nmol/L [221,222]. The results of the RCTs study conducted by O’Callaghan et al. have suggested that this level of 25(OH)D in umbilical blood corresponds to a concentration of 25(OH)D in the mother > 50 nmol/L. These concentrations can be achieved by Vit D supplementation at a daily dose of 1200 IU (30 μg/day) [223]. It has been shown that although the supply of 400 IU (10 μg) of vitamin D per day should prevent a decrease in the concentrations of 25(OH)D < 30 nmol/L, higher doses of 1000 IU (25 μg) may be necessary to ensure the concentration of 25(OH)D in umbilical blood > 30 nmol/L [224].

So far, no dose of this vitamin has been established to achieve effects except the skeletal system. A daily dose of 200–400 IU is widely recommended, but it may be highly insufficient. The results of many studies have indicated that only 4000 IU per day used for 2–3 months allows achieving a 25(OH)D concentration > 75 nmol/L in the mother. The daily dose of 400–600 IU of Vit D, which is contained in the prenatal vitamin sets recommended by the IOM, is insufficient for pregnant women with Vit D deficiency and/or limited sun exposure [225].

There are a few studies on Vit D safety in pregnant women. It has been reported that its dose of up to 4000 IU provided to pregnant women from the 12th to 16th week of pregnancy until delivery seemed to be safe, with no reported cases of hypercalcemia or hypercalciuria [226]. Current evidence supports the concept that circulating 25-hydroxyvitamin D levels during pregnancy should rather be 100–150 nmol/L (40–60 ng/mL), suggesting either a high-dose daily intake of 4000 IU or high-dosage interval bolus (35,000 IU/week or more) to attain this serum levels [227,228].

However, so far, no safe level of Vit D in pregnancy and a safe daily dose have been determined [121]. Its excessive supply in pregnancy can be potentially dangerous for the fetus. This issue was studied in an animal model, which showed that exposure to excessive doses of vitamin D is associated with a high risk of developing supravalvular aortic stenosis [229].

## 5. Vitamin D and Preeclampsia—Experimental Research 

With regard to the multiple mechanisms of action of Vit D, its deficiency seems to be one of the possible factors conducive to PE development, which is confirmed by many reports [230]. Studies conducted by Baca et al. have shown associations between allelic variation in Vit D metabolism genes and PE [231]. It has been suggested that the consequence of low Vit D levels may be the appearance of an early, severe form of PE, and its supplementation may be a protective factor against its recurrence in subsequent pregnancies [232].

The relationship between Vit D and PE development may explain its impact on implantation, angiogenesis, and endothelial status, regulation of the immune response, effect on RAAS, and calcium metabolism.

The main theoretical basis for the use of Vit D in the prevention of preeclampsia is presented in Figure 2.

### 5.1. Trophoblast

The potential Vit D contribution in placentation has been suggested [233]. However, the exact role of vitamin D in this process has still not been settled.

It has been shown that 1,25(OH)2D affects the expression of the HOXA10 gene which is responsible for the implantation and trophoblast invasion into the decidua [234]. A beneficial effect of Vit D on pregnancy development could be observed only if supplementation is initiated during placental implantation [235]. Studies by Barrer D et al. have revealed that Vit D indirectly by intensifying the synthesis of progesterone and human chorionic gonadotrophin (hCG) may improve trophoblast implantation [236]. Although human decidual cells at the fetal-maternal interface synthesize 1,25(OH)2D via CYP72B1 [237], however it has been observed that cultured syncytiotrophoblast cells from preeclamptic placentas have only one-tenth activity of this enzyme compared to the normal cells [238].

The molecular mechanisms explaining the Vit D effect on EVT cells’ migratory and invasive properties are not fully understood. Vitamin D has been shown to regulate the actin cytoskeleton in trophoblast cells. Results of in vitro studies conducted by Chan et al. have suggested that under the influence of 1,25(OH)2D or 25(OH) there is a significant improvement in the invasion of human EVT. They have confirmed the role of Vit D and indicated that its appropriate level could improve this process, and thus, it may constitute one of the protective elements against the PE development [239]. CYP27B1, VDR, VDBP, 25-hydroxylase, and 24-hydroxylase expression has been found in syncytial trophoblasts responsible for invasion [238]. The balance between these enzymes is significantly disturbed in the placental tissue from patients with PE. In preeclamptic placentas, increased expression of CYP27B1, CYP24A1 and reduced CYP2R1 and VDR 25-hydroxylase have been demonstrated compared to healthy placentas, indicating impaired Vit D metabolism in preeclampsia. In addition, the presence of a hypoxic-inducing agent responsible for the development of oxidative stress was found in preeclamptic placental tissue. It has been shown that in placentas derived from healthy women under its influence, changes similar to those observed in preeclamptic placentas occur [240].

Zabul et al. have pointed to the potential significance of an adequate placental concentration of 1,25(OH)2D in PE prevention. They believe that calcitriol by competitive inhibition of placental cytochrome P450scc restrains the excessive synthesis of lipid peroxides and progesterone promoting PE development [241].

The process of trophoblast implantation requires the destruction of the extracellular matrix, for which metalloproteinases are responsible. It has been shown that the reduced levels of vascular MMP-2 and MMP-9 are responsible for vasoconstriction and, as a result, lead to the development of GH and PE [63]. Results of research conducted by Ganguly et al. have indicated that Vit D by enhancing the expression of MMP-2 and MMP-9 promotes the migration and invasion of human EVT in the 1st trimester of pregnancy [234].

### 5.2. Angiogenic Factors and Endothelium 

Vitamin D significantly affects blood vessels and angiogenesis. It is postulated that it may play a beneficial role in preventing endothelial damage and controlling blood pressure in pregnant women with preeclampsia [242]. Under the Vit D influence, the activation of endothelium cells caused by cytokines is limited as well as TNF-α-induced expression of adhesive molecules [243,244]. The results of the Shulz et al. study have shown that gene expression for anti-angiogenic factor (sFlt-1) and surprisingly, pro-angiogenic factor (VEGF) was significantly inhibited at a 25(OH)D concentration ≥ 100 ng/mL compared to the lower 25(OH)D levels. These authors believe that adequate Vit D supplementation ensuring this 25(OH)D level may reduce the risk of PE development [245]. However, most studies have indicated that vitamin D upregulates VEGF gene expressions [246,247,248]. Grundmann et al. have observed that by increasing VEGF expression and pro-matrix metalloproteinase (pro-MMP-2) activity, Vit D induces angiogenesis in endothelial progenitor cells [164]. It has been found that by restoring functional properties of endothelial colony-forming cells (ECFC), which are endothelial progenitor cells, and participate in vasculogenesis and endothelial repair, Vit D may reduce the severity of PE symptoms resulting from endothelial damage [249]. Brodowski et al. have also confirmed the beneficial 1,25(OH)2D influence on endothelial progenitor cells, which allows reversing endothelial damage characteristic of PE [250].

### 5.3. Immune System 

The immunomodulatory properties of Vit D may explain its favorable effect on reducing the risk of PE development [251]. Vitamin D limits the overexpression of Th1, which is characteristic of placentas in preeclampsia [252]. Expression of pro-inflammatory cytokines such as TNF-α and IL-6 was inhibited in placental tissues collected from patients with PE and treated with 1,25(OH)2D compared to trophoblast cell cultures without 1,25(OH)2D [253].

The results of studies among women with PE have shown that compared to healthy ones, they were characterized by significantly lower Vit D levels and elevated levels of IL-6, although no correlation was observed between their concentrations [254].

It has been suggested that it also regulates the proper response of the maternal immune system to the placenta, which prevents the release of anti-angiogenic factors [255].

### 5.4. RAAS

Although the ultimate role of RAAS in the development of PE has not been clearly defined, it has been shown that ATR1-AA are responsible for the development of hypertension [72]. In an animal model, it has been demonstrated that the Vit D administration significantly reduces the blood pressure induced by ATR1-AA [71].

## 6. Vitamin D and Preeclampsia Risk

Due to the multitude of functions of vitamin D, especially its immunomodulatory properties and its beneficial effect on angiogenesis and vascular endothelium, its use in preventing preeclampsia seems attractive. The results of several experimental, clinical, observational, and randomized studies and meta-analyses on this issue have been published. Searching the PubMed database using the keywords “vitamin D” and “preeclampsia” only from the last ten years gives the result of 360 articles. However, only a tiny percentage of them attempted to answer whether Vit D can effectively prevent PE.

This chapter presents the results of randomized controlled trials and meta-analyses that have been published over the past ten years. Electronic databases PubMed has been searched using keywords such as “Vitamin D” and “preeclampsia”. Only articles available in English were considered. Only 5 out of 14 published RCTs and 16 out of 30 meta-analyses provided information on the effect of vitamin D on preeclampsia.

The results of the selected RCTs, which have been released within the last ten years and present the information on the Vit D influence on PE risk, are presented in Table 3.

The presented papers assessed not only the effectiveness of Vit D administration in PE prevention, but their authors also analyzed the relationship of Vit D concentration with the risk of PE [217,228,256,257,258], and the legitimacy of Vit D deficiency screening to prevent PE [256]. Additionally, studies by Mirzakhani et al. determined the expression of 348 Vit D-dependent genes in preeclamptic patients [217].

According to the results of the presented RCTs, it might be concluded that Vit D supplementation seems ineffective in the prevention of PE. Mirzakhani et al., Sablok et al. and Karamali et al. have not demonstrated a beneficial effect of Vit D supplementation in reducing the risk of PE, even despite the inclusion of this treatment in the second trimester of pregnancy (Mirzakhani et al.) [217,228,257]. In contrast, in Sablok et al. and Karamali et al. studies, Vit D was offered late, between the 20th–32nd week [228,257]. For these cases, the late start of Vit D administration and the lack of adequate concentration during placental implantation seem to have influenced the results. Although the difference in doses used in these studies is significant 4400 vs. 50,000 IU, and different administration regimens have been used, it does not affect the results. In contrast, the results of the study conducted by Ali et al. have indicated the effectiveness of the 4000 IU dose in preventing PE. In this research, all patients were started with Vit D supplementation at week 13, which may have significantly affected the outcome [258]. The group studied by Karamali et al., although consisting of patients at high risk of developing PE, was tiny. Hence the results of this study may not be representative [257]. It is also noteworthy that the very wide range of doses from 4400 IU per day [217] through 50,000 [256,257] to even 300,000 [256] was provided at various time intervals in the studied groups. So far, no organization recommends such high doses of vitamin D (50,000 and 300,000 IU) during pregnancy. Low doses of vitamin D (400 IU) [223] or no treatment was offered to pregnant women in control groups [25,228,256,257].

However, research by Marzakhani et al. has also yielded promising results. It has been shown that the satisfying level of Vit D defined by the authors as >30 ng/mL, observed in both early and late pregnancy, is associated with a significantly lower risk of PE (RR 0.28, 95 CI: 0.10–0.96) [217]. Results of research conducted by Rostami et al. on a large group of patients have shown that screening of Vit D status in pregnant women from the low-risk group allows reducing the risk of PE by 60% (RR 0.40, 95% CI: 0.30–0.60. The NNS (numbers needed to screen) value has been estimated at 11 (95% CI, 8 to 17), which means that screening 11 pregnant women will prevent 1 case of PE. In this study, patients subjected to screening were divided into subgroups depending on the concentration of Vit D—the level of >20 ng/mL was considered sufficient, and the pregnant women did not receive Vit D (control group). The study group had a concentration of Vit D < 20 ng/mL, and a subgroup of moderate (10–20 ng/mL), and severe deficit (<10 ng/mL) has been separated. A dose of 300,000 IU given once to women with moderate and 300,000 IU given twice in women with severe Vit D deficiency and maintenance dose of 50,000 IU 1x per month have been shown to be effective in achieving > 20 ng/mL in the perinatal period (RR 1.7, 95% CI:1.2–24 and RR 2.3, 95% CI: 1.7–3.3, respectively). The authors did not note the significant side effects of such high doses of Vit D in pregnant women, but their effects on offspring were not evaluated. These authors believe that the results of their study raise the question of the effective dose of Vit D in preventing pregnancy complications and suggest that this dose may be significantly higher than the currently proposed [256].

The presented research results have suggested that the critical issue in assessing the role of Vit D in preventing PE may not be its dose but appropriate serum concentration. Vitamin D in the dose according to the serum levels needs to be offered in the 1st trimester and even in the periconception period. However, so far, the optimal concentration of Vit D in pregnancy has not been determined.

The results of the selected meta-analysis on the Vit D influence on PE prevalence published within the last ten years are presented in Table 4.

The results of the meta-analyses published in the last ten years also do not allow determining unequivocally whether Vit D is effective in preventing PE. The presented meta-analyses are not homogeneous: some refer to studies assessing the impact of its administration on obstetric results [218,220,259,260,261,262,263,264,265,266,270], the others determine its protective level against the development of PE and other complications of pregnancy [230,264,266,268,269,271]. Meta-analyses were based on studies in the general population of pregnant women without specifically separating groups at high risk of developing PE.

According to the results of most of the meta-analyses presented, the administration of Vit D allows diminishing the risk of PE significantly [218,220,259,260,265,266,270]. Some studies, however, have not shown such a relationship [261,262,263]. Only Gallo et al. assessed the effect of Vit D on PE and GH separately [261]. Other researchers have referred only to preeclampsia.

The inconclusive results of meta-analyses can be explained by the fact that different types of studies were considered in which different doses of vitamin D were analyzed, administered in different time patterns in women with different Vit D concentrations. Similarly, the gestational age when Vit D was offered to pregnant women was heterogeneous and included all trimesters. The authors of the meta-analyses also included in the evaluation studies in which PE was diagnosed based on heterogeneous criteria [230,270,271].

In contrast, the results of meta-analyses assessing the relationship of vitamin D concentrations with the risk of preeclampsia has indicated that its low levels are associated with an increased risk of PE. Higher levels seem to provide protection against PE development [230,265,267,268,271] with the cut-off points used for Vit D concentrations being as follows: <20 ng/mL, <50 ng/mL, and <75 ng/mL. Aghajafari et al. in their meta-analysis, used two discriminators: level 25(OH)D < 50 nmol/L and <75 nmol/L, while the third group was defined as insufficient 25(OH)D levels. They determined the level of <75 nnmol/L for this group and included studies that reported outcomes as proportions of two cut-off categories sufficient and insufficient [267]. A meta-analysis of Hypponen et al. has evaluated the effect of higher serum 25(OH)D levels on the risk of developing PE without specifying a value of this level. The term was used as defined in each study eligible to their meta-analysis [266]. Only one of the presented analyses by Martínez-Domínguez et al. has indicated that the concentration of Vit D does not affect the risk of PE development [269].

The meta-analysis by Aguilar-Cordero et al. has presented random and fixed effects of meta-analyses of observational studies, which differ significantly. Fixed effects are auspicious and indicate a significant effect of low 25(OH)D concentration on PE risk, while random effects do not confirm such a relationship. The authors decided to make this assessment due to the high heterogeneity of the studies included in the meta-analysis [264].

The authors of the presented meta-analyses have indicated the high heterogeneity of the included studies concerning the dose and type of Vit D supplementation and the duration of its use.

## 7. Conclusions

Currently, the commonly recommended preeclampsia prophylaxis is the use of low doses of acetylsalicylic acid in high-risk pregnant women and, by some institutions, calcium supplementation in groups with its deficiencies in the diet. Numerous studies are being conducted on the use of other substances and drugs for this purpose, which, due to their properties and mechanisms of action, could prevent the development of preeclampsia. One of these thoroughly studied substances is vitamin D. Based on the results of research explaining its mechanism of action and understanding the reasons and pathophysiology of the development of preeclampsia, it might be postulated that an anti-inflammatory effect of Vit D and its beneficial influence on the endothelium constitutes its potential use in PE prevention. Unfortunately, the results of the randomized controlled trials and meta-analyses are ambiguous. Despite the multitude of studies published on this subject, there are no clear conclusions about its effectiveness in PE prevention which could form the basis for developing universal recommendations.

In the light of the available data, the following issues regarding the role of vitamin D in preventing preeclampsia remain unresolved: 1. Should the target group for vitamin D supplementation be all pregnant women or only those at high risk? 2. Should it be recommended to test the vitamin D concentration in the periconceptional period and the early first trimester in all pregnant women or a high-risk group? 3. Since when (during the planning period of pregnancy or in the first trimester), how long and what doses of vitamin D should be proposed considering the safety of offspring? 4. What vitamin D level in the periconceptional period and the first trimester should be considered sufficient? These questions may determine the direction of research on vitamin D in the prevention of preeclampsia.

It seems that the fundamental issue, despite the extensive literature, remains the assessment of Vit D concentration in the periconceptional period and/or the early first trimester and defining levels that would allow reducing the risk of PE development. Women with risk factors for Vit D deficiency such as obesity, kidney, liver, thyroid gland diseases, chronic bowel diseases, autoimmune diseases, asthma, diabetes t.2, hypertension, and chronic glucocorticoids, antiepileptic and antiretroviral drug treatment would benefit the most from screening. It appears that patients with risk factors for PE development and Vit D deficiency may require higher doses of vitamin D than commonly recommended for pregnant women.

## Figures and Tables

**Figure 1 nutrients-13-03854-f001:**
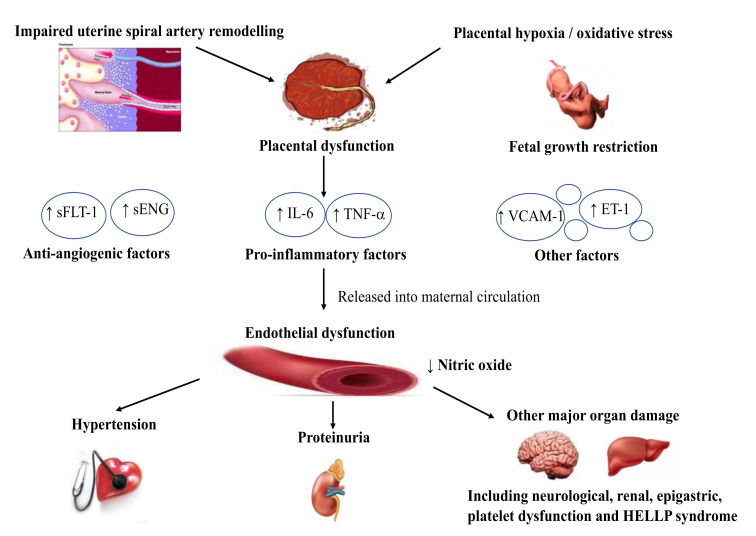
Main stages in PE pathogenesis. sEnd—soluble endoglin, sFlt-1—fms-like tyrosine kinase-1, VICAM1—vascular cell adhesion molecule 1, IL-6—interleukin 6, TNFα—tumor necrosis factor α, ET-1—endothelin-1, HELLP—hemolysis, elevated liver enzymes, low platelets count.

**Figure 2 nutrients-13-03854-f002:**
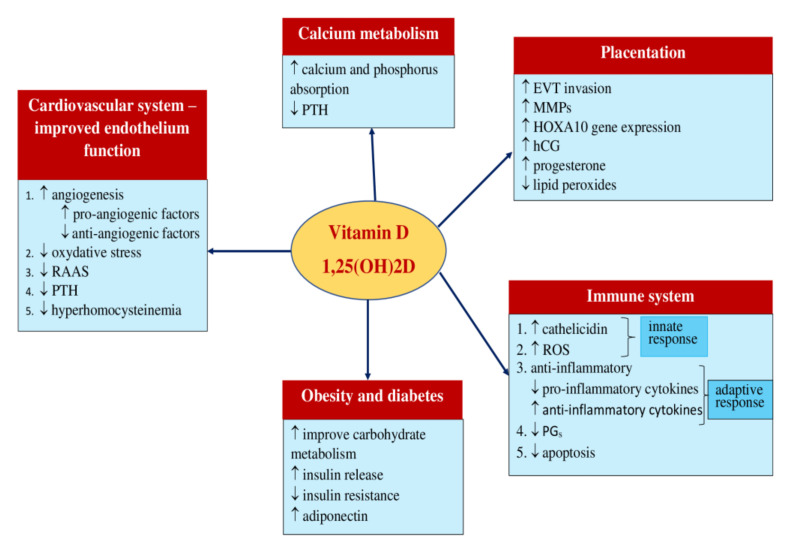
Theoretical basis for the use of Vit D in the prevention of preeclampsia. 1,25(OH)2D - 1,25-dihydroyxvitamin D, PTH – parathyroid hormone, RAAS - renin-angiotensin-aldosterone system, EVT – extravillous trophoblast, MMPs - metalloproteinases, hCG – human chorionic gonadotropin, ROS - reactive oxygen species, PGs - prostaglandins.

**Table 1 nutrients-13-03854-t001:** Vitamin D deficiency in the general population—definitions.

Appropriate Vit D Level	≥50 nmol/L
Vit D deficiency—mild	30–49 nmol/L
Vit D deficiency—moderate	12.5–29 nmol/L
Vit D deficiency—severe	<12.5 nmol/L

**Table 2 nutrients-13-03854-t002:** Recommendations on vitamin D prophylaxis during pregnancy.

	Recommended Daily Dose of Vitamin D (IU)	Minimal Vit D (25(OH)D) Level (nmol/L)
WHO	200	>50
Institute of Medicine (USA)	600–1000	≥30
Endocrine Society (USA)	1500–2000	≥75
ACOG (USA)	600	≥50
NICE (UK)	400–800	>30
RANZCOG	400–2000	>50
PTGiP	1500–2000 BMI > 30 kg/m^2^–up to 4000	No data

WHO—World Health Organization. ACOG—American College of Obstetricians and Gynecologists. NICE—National Institute of Health and Care Excellence. RANZCOG—Royal Australian and New Zealand College of Obstetricians and Gynecologists. PTGiP—Polish Society of Gynecologists and Obstetricians.

**Table 3 nutrients-13-03854-t003:** Selected randomized placebo-controlled trials on vitamin D influence on PE risk.

Author	Aim of the Study	Size of Groups	Vit D Dose (IU) and Duration of Treatment	GA at the Entry to the Study	Main Outcome
Mirzakhani et al. 2016 [217]	PE risk	Vit D (SG) 408 CG 408	4400 daily 400 daily	10–18th week	PE incidence SG 8.08% CG 8.33%, NS RR 0.97 95% CI: 0.61–1.53
Rostami et al. 2018 [256]	Vit D status screening	Screened Vit D 800 Without Vit D 200 Non screened 900	50,000–300,000 weekly or monthly; 6–12 weeks	<14th week	Screening reduces PE risk by 60% RR 0.40 95% CI: 0.30–0.60
Karamali et al. 2015 [257]	PE risk	Vit D (SG) 30 CG 30 patients with high PE risk	50,000 every 2 weeks	20–32nd week	PE incidence SG 3.3% CG 10% *p* = 0.3
Sablok et al. 2015 [228]	Pregnancy complication risk	Vit D (SG) 120 CG 60	60,000–120,000 every 4 weeks	20–32nd week	PE incidence SG 11.1% CG 21.1% *p* = 0.08
Ali et al. 2019 [258]	PE risk	Vit D (SG) 83 CG 81	4000 daily	at 13th week up to 12th week after delivery	PE incidence SG 1.2% CG 7.4% *p* = 0.049

PE—preeclampsia; Vit D—vitamin D; SG—study group; CG—control group; GA—gestational age; P—statistical significance; RR—relative risk; CI—confidence interval.

**Table 4 nutrients-13-03854-t004:** Selected meta-analyses on Vitamin D influence on PE risk.

Authors	Studied Group	Number of Participants	Impact on PE	Additional Information
Khaing et al. 2017 [220]	Vit D vs. placebo	357	RR 0.47 95% CI: 0.24–0.89	NNT 17
Palacios et al. 2016 [259]	Vit D vs. no treatment	219	RR 0.52 95% CI: 0.25–1.05	PE occurrence 8.9% vs. 15.5%
Palacios et al. 2019 [218]	Vit D vs. no treatment	499	RR 0.48 95% CI: 0.30–0.79	
Fogacci et al. 2020 [260]	Vit D vs. no treatment Vit D vs. no treatment < 20th week	4777	RR 0.37 95% CI: 0.26–0.52 RR 0.35 95% CI: 0.24–0.50, *p* < 0.001	Increasing dose–decreasing PE risk RR −1.10 95% CI: −1.73–1.46, *p* < 0.001
Gallo et al. 2020 [261]	Vit D vs. no treatment	364	PE RR 0.7 95% CI: 0.4–1.4, NS GH RR 0.8 95% CI: 0.3–2.2, NS	
Pérez-López et al. 2015 [262]	Vit D vs. placebo	877	RR 0.88 95% CI: 0.51–1.52, NS	
Roth et al. 2017 [263]	Vit D vs. no treatment	3398	RR 1.09 95% CI:0.43–2.76, NS	
Aguilar-Cordero et al. 2020 [264]	Random effects meta-analysis 25(OH)D < 75 nmol/L 25(OH)D < 50 nmol/L Fixed effect meta-analysis 25(OH)D < 75 nmol/L 25(OH)D < 50 nmol/L Interventional studies Vit D supplementation	10,979 14,496 10,979 14,469 1660	RR 1.26 95% CI: 0.87–1.82, NS RR 1.42 95% CI: 0.99–2.04, NS RR 1.44 95% CI: 1.26–1.64 *p* < 0.00001 RR 1.47 95% CI: 1.29–1.67 *p* < 0.00001 RR 0.68 95% CI: 0.49–0.95	
Akbari et al. 2020 [230]	25(OH)D < 20 ng/ml	21,546	Fixed RR 1.33; *p* < 0.0001; random RR 1.54 *p* = 0.0029	
Fu et al. 2018 [265]	Vit D supplementation	21,127	RR = 0.41 95%CI = 0.22-0.78	
Hyppönen et al. 2014 [266]	Vit D supplementation early in pregnancy higher serum 25(OH)D Vit D supplementation	59,789 5058 5982	RR 0.81 95% CI: 0.75–0.87 *p* < 0.000001 RR 0.52 95% CI: 0.30–0.89 *p* = 0.02 RR 0.66 95% CI: 0.52–0.83 *p* = 0.001	
Aghajafari et al. 2013 [267]	Observational study Insufficient 25(OH)D levels 25(OH)D < 75 nmol/L 25(OH)D < 50 nmol/L	3190	RR 1.79 95% CI: 1.25–2.58 RR 2.11 95% CI: 1.36–3.27 RR 1.27 95% CI: 0.60–2.42	
Tabesh et al. 2013 [268]	Vit D deficiency 25(OH)D ≤ 50 nmol/L (20 ng/mL), 25(OH)D <38 nmol/L (15.2 ng/mL)	1775 931	RR 2.78 95% CI: 1.45–5.33 NS	
Martínez-Domínguez et al. 2018 [269]	First half of pregnancy normal 25(OH)D (≥30.0 ng/mL) Insufficient (20.0–29.9 ng/mL) Deficient (<20.0 ng/mL)	817 323 494	RR 0.73 95% CI: 0.35–1.51, NS RR 0.79 95% CI: 0.28–2.21, NS RR 0.67 95% CI: 0.24–1.89, NS	
Kinshella et al. 2021 [270]	Vit D supplementation	1353	RR 0.62 95% CI: 0.43–0.91 NS	Decrease in PE risk by 38%
Yuan et al. 2021 [271]	Low 25(OH)D levels	39,031	RR 1.62 95% CI: 1.36–1.94 *p* < 0.001	

Vit D—vitamin D; 25(OH)D—25-hydroxyvitamin D; NNT—numbers needed to treat; P—statistical significance.; RR—relative risk; CI—confidence interval.

## Data Availability

MDPI Research Data Policies.

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
