# Peer review of "Could Vitamin D Be Effective in Prevention of Preeclampsia?"

_nutrients, 2021, doi:10.3390/nu13113854_

Round 1

Reviewer 1 Report

This review titled ‘Could vitamin D be effective in prevention of preeclampsia?’ explores the use of prophylaxis vitamin D.

Whilst prevention of PE is an active area of research, the use of vitamin D has shown little promise by a number of independent reviews and meta-analyses. This reduces the interest in this topic slightly. However if restructured there are interesting theories to be explored in this space.

 The structure of the article needs to be improved dramatically to increase impact. There is a large section of preeclampsia followed by a large section on Vitamin D. In its current form it also includes a lot of irrelevant information. Mechanisms of both preeclampsia and vitamin D are discussed that do not relate to how vitamin D may be effective in preventing preeclampsia. Only discuss relevant information to vitamin D use.

The entire document would benefit from revision of grammar and sentence structure. Be sure to define abbreviations on first use and then continue to use the abbreviated version. Do not switch between the full and abbreviated version. For example, preeclampsia and PE.

The conclusions raise additional questions rather than conclude what was determined from the current literature. I would rephrase this section.

Author Response

On behalf of the authors, I would like to thank the Reviewer very much for the thorough evaluation of the manuscript and suggestions that will improve our article. We do appreciate the commitment and work of the Reviewer on improving the quality of our article.

Please let me refer to the comments of the Reviewer.

This review titled ‘Could vitamin D be effective in prevention of preeclampsia?’ explores the use of prophylaxis vitamin D.

Whilst prevention of PE is an active area of research, the use of vitamin D has shown little promise by a number of independent reviews and meta-analyses. This reduces the interest in this topic slightly. However, if restructured there are interesting theories to be explored in this space.

The structure of the article needs to be improved dramatically to increase impact. There is a large section of preeclampsia followed by a large section on Vitamin D. In its current form it also includes a lot of irrelevant information. Mechanisms of both preeclampsia and vitamin D are discussed that do not relate to how vitamin D may be effective in preventing preeclampsia. Only discuss relevant information to vitamin D use.

The pathogenesis of preeclampsia has been presented in detail. Those elements in the development of PE that can be affected by vitamin D have been highlighted. Hence, the part on the mechanisms of action of vitamin D is extensive. Since the journal "Nutrients" is multidisciplinary, we believe that the chapter on PE pathogenesis will be of interest to specialists in nutrition and the presentation of vitamin D metabolism and the mechanisms of its action – for gynecologists and obstetricians. Both chapters have been reviewed, and irrelevant information has been removed. The relationship between Vit D and PE development may explain its impact on implantation processes, angiogenesis and endothelial status, regulation of the immune response, effect on RAAS and calcium metabolism. Hence, the theoretical basis for the action of vitamin D as an agent for preventing PE was presented in the chapter "Vitamin D and preeclampsia – experimental research”.

The entire document would benefit from revision of grammar and sentence structure. Be sure to define abbreviations on first use and then continue to use the abbreviated version. Do not switch between the full and abbreviated version. For example, preeclampsia and PE.

The manuscript has been thoroughly reviewed and corrected by a native speaker, and suggested changes have been made wherever possible.

The conclusions raise additional questions rather than conclude what was determined from the current literature. I would rephrase this section.

The questions in the "Conclusions" chapter have arisen after a thorough assessment of the current literature on the effectiveness of vitamin D in the prevention of PE and, in our opinion, may justify further research on this issue despite several published works on this subject. The chapter is supplemented with the sentence "It appears that patients with risk factors for PE development and vitamin D deficiency may require higher doses of vitamin D than commonly recommended in pregnancy." 

Reviewer 2 Report

The authors have presented a detailed and useful description of the pathogenesis of preeclampsia. 

For vitamin D the authors acknowledge that “the data on the benefits of its supplementation to reduce the risk of PE are inconclusive”, but throughout the paper the major focus seems to be on the benefits of vitamin D supplementation.  The paper will be more useful if both positive and negative results are thoroughly reviewed. 

The word “proven” is used many times throughout the paper which may be a language problem rather than an inadequate understanding of statistics.  Also, the authors may want to avoid using phrases like “It is believed that…..”.  

Structure of the paper is difficult to follow.  Under 3.2  Mechanism of action [referring to vitamin D] there are seven subsections, all labelled “a”.  Additionally, the level of scientific consensus for a role of vitamin D for each of these subsections is highly variable. 

Furthermore, the paper seems to have lost focus on preeclampsia, particularly under section 3.2.  For example, authors need to explain more clearly the value added by  subsections on obesity and diabetes and on cancer. 

The title for Table 1 “Degree of vitamin D deficiency in general population” doesn’t seem to be a good description for the contents of the table  

Throughout the paper, the authors mention that results of RCTs and meta-analyses are ambiguous.  But in several places the text then emphasizes the results of papers supporting roles for vitamin D.

In several places, large supplemental doses are mentioned.  The Institute of Medicine (USA) did establish a tolerable upper level for vitamin D intake of 4000 IU (100 µg).  Several reports of vitamin D toxicity have been published and risks of toxicity need to be more thoroughly acknowledged in the article.

It has been demonstrated that a higher percentage of inadequate circulating levels of 25(OH) D are observed in several population groups, such as dark skinned individuals, and Middle Eastern women. Are there data showing that rates of preeclampsia are higher in these groups?

Editorial Issues

  1. 633 - Check “groving” body of evidence
  2. 902 – Check “level of > 20 g/mL was considered sufficient”

Author Response

On behalf of the authors, I would like to thank the Reviewer very much for the thorough evaluation of the manuscript and suggestions that will improve our article. We do appreciate the commitment and work of the Reviewer on improving the quality of our article.

Please let me refer to the comments of the Reviewer.

The authors have presented a detailed and useful description of the pathogenesis of preeclampsia. 

For vitamin D the authors acknowledge that “the data on the benefits of its supplementation to reduce the risk of PE are inconclusive”, but throughout the paper the major focus seems to be on the benefits of vitamin D supplementation.  The paper will be more useful if both positive and negative results are thoroughly reviewed. 

Throughout the paper, the authors mention that results of RCTs and meta-analyses are ambiguous.  But in several places the text then emphasizes the results of papers supporting roles for vitamin D.

The chapter "Vitamin D and preeclampsia risk" presents the results of RCTs and meta-analyses that assessed two issues: 1. Effectiveness of vitamin D in the prevention of preeclampsia, 2. Relationship between vitamin D concentration and the occurrence of PE. The results of these studies are presented in Tables 4 and 5 with an overview. These results do not clearly indicate the effectiveness of vitamin D in preventing preeclampsia, and such a statement was included in the manuscript. In our opinion, only the results of those studies that link the low concentration of vitamin D with the PE occurrence were emphasized because most of them confirm the existence of such a relationship. The results of studies that do not confirm such a link were also discussed. 

The word “proven” is used many times throughout the paper which may be a language problem rather than an inadequate understanding of statistics.  Also, the authors may want to avoid using phrases like “It is believed that…..”.  

The proposed language changes have been made.

Structure of the paper is difficult to follow.  Under 3.2  Mechanism of action [referring to vitamin D] there are seven subsections, all labelled “a”.  Additionally, the level of scientific consensus for a role of vitamin D for each of these subsections is highly variable. 

Furthermore, the paper seems to have lost focus on preeclampsia, particularly under section 3.2.  For example, authors need to explain more clearly the value added by  subsections on obesity and diabetes and on cancer. 

Chapter 3.2 aims to present the most important mechanisms of action of vitamin D that may justify the potential consideration of its use in the prevention of preeclampsia. The number of subsections has been changed and their order has been improved. After considering the legitimacy of diabetes and cancer subsections, we decided to remove them.

The title for Table 1 “Degree of vitamin D deficiency in general population” doesn’t seem to be a good description for the contents of the table.

The title has been changed.  

In several places, large supplemental doses are mentioned.  The Institute of Medicine (USA) did establish a tolerable upper level for vitamin D intake of 4000 IU (100 µg).  Several reports of vitamin D toxicity have been published and risks of toxicity need to be more thoroughly acknowledged in the article.

The article has been supplemented with information on the recommended doses of vitamin D and its toxicity.

It has been demonstrated that a higher percentage of inadequate circulating levels of 25(OH) D are observed in several population groups, such as dark-skinned individuals, and Middle Eastern women. Are there data showing that rates of preeclampsia are higher in these groups?

“In women who have lower levels of vitamin D, i.e. in dark-skinned and Muslim women, there is a higher incidence of preeclampsia.” This information has been included in the text.

Editorial Issues

  1. 633 - Check “groving” body of evidence
  2. 902 – Check “level of > 20 g/mL was considered sufficient”

These mistakes have been corrected.

Reviewer 3 Report

Thank you for asking me to review this interesting paper. I have some suggestions that may improve the clarity of the paper:

Introduction,Line 58; Do you mean 'delivery of the baby'.?

Pathogenesis of PE: This section is rather long and there is a lot of detail here. Including some figures/flow charts may make it easier for the reader to understand some of these mechanisms and reduce the wordiness of this section.

There is no mention of MTHF reductase gene mutation; this can cause PE and is also linked to micronutrient intake (Folate & Riboflavin), so worth mentioning here.

Vitamin D: The prevalence of vitamin D deficiency is reported to be increasing in women & children in northern European countries, may be as a result of over caution with sun screen and concerns about skin cancer. This is worth a mention too.

Please also ensure that your phrases use a 'patient first approach', referring to people living with obesity (rather than obese people).

Recommendations on supplementation of vitamin D

Many European countries already recommend routine supplementation of vitamin D during pregnancy (including the UK at 10µg/day), but not necessarily to reduce risk of PE. This needs to be highlighted - plus is there any evidence that this routine supplementation has had any impact of PE prevalence in countries where this is recommended?

Conclusion

Here you discuss whether supplementation should be routine for all pregnant women, or just targeted at those with increased risk of PE - however many European countries already recommend routine vitamin D supplementation. In the UK this recommendation has existed since 2010, so this is not new. Could it be that the women at increased risk of PE and/or vitamin D deficiency may need a higher dose of vitamin D (> 10 µg/day)?

Author Response

On behalf of the authors, I would like to thank the Reviewer very much for the thorough evaluation of the manuscript and suggestions that will improve our article. We do appreciate the commitment and work of the Reviewer on improving the quality of our article.

Please let me refer to the comments of the Reviewer.

Introduction,Line 58; Do you mean 'delivery of the baby'.?

Indeed, it is about the delivery of the baby, but in this context, delivery means the end of pregnancy. The very phrase “delivery” is used in this meaning in obstetrics textbooks, statements, recommendations of scientific societies, e.g. quoting from ACOG: “Therefore, delivery is recommended when gestational hypertension or preeclampsia with severe features (Box 3) is diagnosed at or beyond 34 0/7 weeks of gestation, after maternal stabilization or with labor or prelabor rupture of membranes. Delivery should not be delayed for the administration of steroids in the late preterm period.” (ACOG Practice Bulletin No. 202: Gestational Hypertension and Preeclampsia. Obstet Gynecol. 2019; 133: e1-e25. doi: 10.1097/AOG.0000000000003018.)

Pathogenesis of PE: This section is rather long and there is a lot of detail here. Including some figures/flow charts may make it easier for the reader to understand some of these mechanisms and reduce the wordiness of this section. There is no mention of MTHF reductase gene mutation; this can cause PE and is also linked to micronutrient intake (Folate & Riboflavin), so worth mentioning here.

The pathogenesis of preeclampsia was presented in detail, paying attention to those elements in the development of PE that may be the target of vitamin D. Our intention was to present the theoretical basis for the use of vitamin D in the prevention of preeclampsia. This chapter was reviewed and fragments that are not associated with the action of vitamin D were removed. In addition, the participation of MTHF reductase gene mutation in the pathogenesis of preeclampsia was considered and a figure showing the pathogenetic elements of preeclampsia was attached.

Vitamin D: The prevalence of vitamin D deficiency is reported to be increasing in women & children in northern European countries, may be as a result of over caution with sunscreen and concerns about skin cancer. This is worth a mention too.

Please also ensure that your phrases use a 'patient first approach', referring to people living with obesity (rather than obese people).

The following information has been included in the text.

In women and children in northern European countries, an increase in the prevalence of vitamin D deficiency is observed due to concern for the development of skin cancer and the widespread use of sunscreens.

 The phrases e.g. obese women or obese patients have been replaced by the terms “women with obesity” or “patients with obesity”.

Recommendations on supplementation of vitamin D

Many European countries already recommend routine supplementation of vitamin D during pregnancy (including the UK at 10µg/day), but not necessarily to reduce risk of PE. This needs to be highlighted - plus is there any evidence that this routine supplementation has had any impact of PE prevalence in countries where this is recommended?

Information on the lack of recommendation for the use of vitamin D in the prevention of preeclampsia was placed in the text. So far, there is no information in the literature on whether vitamin D supplementation commonly recommended in many countries affects the incidence of preeclampsia and such information has also been included in the manuscript.

Conclusion

Here you discuss whether supplementation should be routine for all pregnant women, or just targeted at those with increased risk of PE - however many European countries already recommend routine vitamin D supplementation. In the UK this recommendation has existed since 2010, so this is not new. Could it be that the women at increased risk of PE and/or vitamin D deficiency may need a higher dose of vitamin D (> 10 µg/day)?

This chapter considers only the issue of the legitimacy of the use of vitamin D in PE prevention and the statement "there are no clear conclusions about its effectiveness in PE prevention which could form the basis for the development of universal recommendations" concerns the creation of recommendations for the use of vitamin D in the prophylaxis of preeclampsia.

"It appears that patients with risk factors for PE development and vitamin D deficiency may require higher doses of vitamin D than commonly recommended in pregnancy." This sentence was attached to the main text.

Round 2

Reviewer 2 Report

The manuscript is more clear and my questions have been answered.

Author Response

On behalf of the authors, I would like to thank the Reviewer very much for the very thorough analysis of our article and the comments that have significantly improved its quality.

With regarding comments after the first revision:

The manuscript has been checked and corrected linguistically.